# Systematic Review on the Use of Biosimilars of Trastuzumab in HER2+ Breast Cancer

**DOI:** 10.3390/biomedicines10082045

**Published:** 2022-08-21

**Authors:** Eleni Triantafyllidi, John K. Triantafillidis

**Affiliations:** 1Hellenic Society of Gastrointestinal Oncology, 354, Iera Odos Street, Haidari, 12461 Athens, Greece; 2Metropolitan General Hospital, 15562 Holargos, Greece

**Keywords:** biosimilars, breast cancer, cost, monoclonal antibodies, trastuzumab, treatment

## Abstract

Trastuzumab is a monoclonal antibody used in the treatment of breast cancer in cases where the tumor overexpresses the HER2 receptor, a cell membrane receptor activated by the epidermal growth factor. Intravenous and subcutaneous administration of trastuzumab have comparable clinical and pharmacological characteristics, but trastuzumab biosimilars are currently only available in intravenous form. Trastuzumab biosimilars are ultimately preferred by a proportion of patients, especially in cases where co-administration of other chemotherapeutic agents, such as trastuzumab and tucatinib, a small molecule of tyrosine kinase inhibitor, is required in patients with HER-positive metastatic breast cancer. Oncologists should be well-aware of the advantages of intravenously administered trastuzumab biosimilars over subcutaneous administration, certainly also taking into account the patient’s preferences. Further cost-effectiveness analyses will be very important, along with expectations regarding successful concomitant subcutaneous administration of trastuzumab with other anticancer drugs, such as pertuzumab. This systematic review describes and analyzes the so-far published studies concerning the use of the available trastuzumab biosimilars in HER-positive early and metastatic breast cancer in terms of efficacy, safety, and cost–benefit ratio. An attempt was also made to draw some conclusions and to comment on future needs and perspectives.

## 1. Introduction

Breast cancer (BC) is the world’s most commonly diagnosed cancer in women, with more than 2 million new cases in 2020. Its incidence and mortality have increased in recent decades, mainly due to altered risk factors and better recording and detection. About 80% of patients with BC are over 50 years old. Survival depends on both the stage and the molecular subtype. Infiltrative BC includes tumors that show differences in their clinical presentation, behavior, and morphology. Based on mRNA expression levels, BC can be divided into several molecular subtypes (Luminal A, Luminal B, HER2 positive (HER+), and Basal-like). Molecular subtypes help to determine new therapeutic strategies and to separate patients, thus affecting their therapeutic management. The eighth edition of the TNM classification defines a new staging system of BC that, in addition to the anatomical features of the tumor, also takes into account biological factors.

Treatment of BC involves a combination of different treatment strategies applied in a specific sequence, including surgery, radiotherapy, chemotherapy, hormone therapy, and treatment with biological agents [1]. The way BC is treated today has changed significantly as a consequence of the detailed definition of its molecular characteristics, the immunohistochemical receptors (e.g., ER, PR, HER2 (ERBB2), the Ki-67 cell proliferation index (MKI67)), genomic markers (e.g., BRCA1, BRCA2, and PIK3CA), and immunomarkers (e.g., tumor-infiltrating lymphocytes and PD-L1). New emerging combinations of biomarkers are the basis of increasingly complex diagnostic algorithms. Neoadjuvant combination therapy is the standard of care, especially in HER2+ and triple-negative BC, as well as the basis for the de-escalation of breast surgery and neoadjuvant strategies. Radiation therapy is still among the cornerstones in the treatment of BC. ER-positive tumors are treated for a long period (5–10 years) with hormonotherapy and chemotherapy, based on individual risk assessment. For metastatic BC, standard treatment options include targeted approaches, such as CDK4 and CDK6 inhibitors, PI3K inhibitors, PARP inhibitors, and PD-L1 immunotherapy, depending on the type and molecular profile of the tumor [2]. In general, BC is a disease that requires evaluation and management by a multidisciplinary team.

In this systematic review, we report the clinical data regarding the treatment of patients with HER2+ BC with the use of biosimilars—both those available and those under evaluation—to trastuzumab monoclonal antibodies. Apart from the clinical data, important economic factors are also analyzed focusing on the cost–benefit ratio. Finally, an attempt is made to draw some conclusions and to record some prospects for the future of treatment of patients with BC.

### 1.1. Biosimilars

With the term biosimilar, we describe a biological medicinal product with structural and functional properties, including pharmacokinetics and clinical efficacy, similar to that of an approved biological reference medical product [3,4]. The European Medicines Agency (EMA) as well as the United States Food and Drug Administration agency (FDA) developed biosimilar guidelines requiring biosimilars to demonstrate, in relevant clinical and laboratory trials, similar results to those obtained by the use of the original product. The similarity in structural and functional properties is critical to ensure that there will be no significant difference in the clinical results. Based on this fundamental condition, the similarity assessment is achieved with a gradual approach, starting from the detailed physicochemical and biological characterization of the biosimilar and the reference product. Analytical similarity studies should be based on data with well-defined procedures and criteria concerning the quality characteristics of the reference product that could affect the efficiency, strength, and safety of the product. The specific characteristics of biosimilar drugs according to the EMA and European Commission are shown in Table 1.

The introduction of biosimilars in everyday clinical practice is expected to improve access to anti-HER2 therapy by creating additional treatment options and lower-cost alternatives. Since HER2-targeting drugs can be given for a long period and in combination with other systemic therapies, biosimilars can lead to significant resource savings for healthcare systems in many countries [5]. Currently, all approved trastuzumab biosimilars are for intravenous use only, although the expiration of the patent for subcutaneous trastuzumab (expected in 2030 in the US) will allow the development of subcutaneous biosimilar drugs in the future. The fact is, however, that many questions related to the use of these drugs should be answered, mainly concerning the efficacy, safety, cost–benefit, and comparability of these drugs. In the USA, challenges related to the adoption of biosimilars in therapeutic oncology remain and include a lack of targeted training for providers and patients, efficacy and safety concerns, and several other practical issues [6]. Various international regulatory authorities have adopted and issued formal definitions of what is defined as a biosimilar, which are shown in Table 2.

HER2 is a cell surface protein receptor that promotes cell growth and proliferation as a result of its activation. Briefly, the signaling pathways of HER2 are as follows: Binding of the receptor causes dimerization between receptors of the epidermal growth factor receptor (EGFR) family and the HER2 receptor. Then, homodimers or heterodimers stimulate a series of signaling cascades. Among the various signaling pathways, the phosphatidylinositol 3-kinase (PI3K) and mitogen-activated protein kinase (MAPK) pathways are the two main and most studied pathways that play a central role in tumor proliferation. The whole signal transduction process is divided into three sections: signal input (receptor connection and dimerization), signal processing (series of signal cascades), and signal output (corresponding cellular processes) [8]. In 20–25% of BC cases, HER2 is overexpressed, which results in faster cancer growth and a higher recurrence rate [9]. Its overexpression is associated with a more aggressive disease phenotype, as well as with worse disease-free survival rates (DFSR) and overall survival (OS) compared to HER2-negative patients. Several anti-HER2 agents are currently in clinical use, such as the trastuzumab and pertuzumab monoclonal antibodies, lapatinib, neratinib, and tucatinib micromolecular inhibitors, and the ado-trastuzumab emtansine and trastuzumab antibody–drug complexes. These factors differ significantly in the kind of side effects as well as the approved indications, from neonatal and adjuvant therapy in early disease to the first and last line of treatment in metastatic disease [10].

Trastuzumab is a kappa light chain monoclonal antibody specifically targeting HER2. As a result of its development through humanized mouse parent antibodies, trastuzumab consists of human and mouse genomes. Its mechanism of action is related to the inhibition of the proliferation of cells overexpressing HER2 by inhibiting the MAPK and PI3K/Akt pathways. Another important mechanism is the attraction of immune system cells to tumor regions that overexpress HER2, the so-called antibody-induced cytotoxicity. Upon attachment of the antigen to the monoclonal antibody, the effect is mediated by the Fc fragment binding to the Fcγ (FcγR) IIIa (FcγRIIIa) receptor, also known as CD16a [11]. HER2 is activated by dimerization, which induces activation of the Ras/Raf/activated protein kinase (MAPK), PI3K/Akt, and phospholipase Cγ (PLCγ)/C cell proliferation. The half-life after intravenous and subcutaneous administration of trastuzumab is 10 days. Subcutaneous administration of trastuzumab at a dose of 600 mg results in minimally effective drug levels equivalent to those administered intravenously.

Trastuzumab is a biologic therapy for the treatment of BC and metastatic gastric cancer in HER2-positive patients. The drug was approved by the US FDA in 1998 and the EMA in 2000 as a treatment for metastatic BC (HER2+ MBC). The approval of this humanized monoclonal antibody changed the course of treatment of female patients with HER2+ BC, becoming the standard for neoadjuvant and adjuvant chemotherapy, as well as the treatment of metastatic disease. In the US and the EU, trastuzumab has been approved as a first-line treatment for HER2+ MBC. It is also approved for HER2+ Early BC (HER2+ EBC) as adjunctive therapy after surgery, chemotherapy, or radiotherapy in combination with paclitaxel or docetaxel, in combination with docetaxel and carboplatin, or in combination with adjuvant chemotherapy followed by adjuvant monotherapy.

A large number of clinical trials have shown that trastuzumab significantly improves the overall survival and disease-free period in women with early HER2+ BC. In HER2+ EBC, trastuzumab administered for 1 year significantly improved DFSR and OS compared with no trastuzumab regimens. In metastatic BC, the early addition of trastuzumab to cytotoxic chemotherapy significantly improved both response rates and OS. In relevant studies, the combination of trastuzumab and taxane was identified as the gold standard for first-line treatment in patients with metastatic BC. Subsequent studies have shown that HER2 double-blocking in combination with chemotherapy is superior to trastuzumab-only blockade [12]. In combination with chemotherapy, intravenous administration of both trastuzumab and its biosimilars improves OS and overall response rates (ORR) in patients with HER2+ MBC [13,14,15,16,17]. In the Hannah phase III study, comparable efficacy was observed between intravenous and subcutaneous trastuzumab in patients with HER2+ EBC. The results showed no significant differences in clinical efficacy between intravenous and subcutaneous trastuzumab.

### 1.2. Biosimilars of Trastuzumab

As with most biologics, trastuzumab has a high price compared to traditional cytotoxic chemotherapy. Three products associated with trastuzumab-2 antibody conjugate, namely, ado-trastuzumab emtansine, fam-trastuzumab, as well as subcutaneous trastuzumab/hyaluronidase, have now been approved, thus extending the available treatment options. The approval also of five trastuzumab biosimilars at the end of 2019 promises a significant reduction in treatment costs. However, there are obstacles to their use, such as the uncertainty of the outcome of the transition from the reference drug to the biosimilar, as well as the lack of understanding of biosimilars [18]. Observational studies between 2000 and 2015 in patients with HER2+ MBC showed that 12% in the US, 27–54% in Europe, and 27.1–49.2% of patients in China did not receive trastuzumab or any of the other HER2+ agents as first- and/or subsequent-line treatment for metastatic disease. Moreover, observational studies on HER2+ EBC treatment patterns between 2005 and 2015 indicate that 19.1% to 59.5% of patients in areas of North America, Europe, Australia, New Zealand, and China did not receive (neo)adjuvant therapy with trastuzumab. The data indicate that some subgroups of patients, including elderly patients, patients with HER2+/hormone receptor disease, and women with small and/or negative HER2+ lymph nodes, were less likely to receive anti-HER2 therapy. Barriers to accessing trastuzumab are multifactorial and include issues related to drug financing and the high cost of treatment [19].

The administration of trastuzumab is costly for patients, a fact potentially restricting access and also making some healthcare providers reluctant to prescribe this drug. However, the increasing availability of biosimilars is expected to significantly reduce the costs associated with biomarkers over the next decade. The currently approved biosimilars of trastuzumab are shown in Table 3.

At least nine trastuzumab biosimilars are currently being studied, mainly in phase I studies, which are as follows:HLX02AryoTrust^®^SIBP-01TRASTURELAK-HER2BAT8001BCD-022DMB-3111HD201

## 2. Materials and Methods

This systematic review was performed based on acceptable literature guidelines for writing systematic reviews and meta-analyses [20]. Articles were searched in the MEDLINE database for the period up to January 2022 using the following keywords: “Breast” AND “neoplasms” or “neoplasm” or “cancer” or “cancers” or “carcinoma” or “carcinomas” AND “trastuzumab biosimilars”. Only articles published in the English language were evaluated. The relevant literature was reviewed, and the data of each study were recorded. All prospective and retrospective clinical studies were selected. All studies related to the efficacy and safety of trastuzumab in BC were also selected regardless of the size of the sample. In all studies, the following data were recorded: First author, year of publication, trial phase, number of patients, characteristics of the patient population (first-line treatment, second-line treatment, neoadjuvant treatment, etc.), histological type, median age, ORR including both full response rate and partial response (%), median OS (in months), DFSR, and complications. In cases where multiple, overlapping publications from the same study were identified, the most currently published study was included.

The research revealed a total number of 257 articles. Of these, 221 were excluded for the following reasons: (i) not relevant to the subject of the investigation (157), (ii) review articles (59) including 7 systematic reviews and 5 meta-analyses, and (iii) duplicate publications (5). Therefore, 31 articles (clinical studies) were reviewed, of which 21 were included in the final evaluation. Moreover, a total number of 3695 patients on approved biosimilars and 1022 patients on biosimilars under investigation were analyzed. These data are shown in Figure 1. Reasons for excluded studies of approved trastuzumab biosimilars are listed in Table 4.

Specific reference was made regarding the efficacy and safety data for the five USA- and EU-approved biosimilars (Table 5) as well as for the three biosimilars under investigation (Table 6).

## 3. Results

### 3.1. Results of Clinical Application of Approved Trastuzumab Biosimilars

#### 3.1.1. SB3 (Ontruzant^®^)

SB3 (Ontruzant^®^) (Samsung Bioepis Co., Ltd., Incheon, Korea) is the first biosimilar of the trastuzumab reference antibody approved in the EU (November 2017) and the USA (January 2019). The approval applies to all indications for which reference trastuzumab has been approved, i.e., HER2+ EBC, and HER2+ MBC. SB3 has similar physicochemical and pharmacodynamic properties to those of the trastuzumab reference. The pharmacokinetic biosimilarity of the two drugs has been demonstrated in healthy volunteers and women with early or locally advanced HER2+ BC. SB3 has shown clinical efficacy equivalent to that of trastuzumab in women with early or locally advanced HER2+ BC. Even the tolerance, immunogenicity, and safety profile of SB3 are similar to that of reference trastuzumab. The similarity of SB3 with the reference product was proved through analytical characterization of the quality profile determined during the development period of the drug. The analytical similarity of SB3 with the reference drug was also evaluated with qualitative characteristics covering a wide range of structural/physicochemical and functional parameters. SB3 is similar to the reference product in terms of its biological quality characteristics [21].

In a relevant study, the similarity of SB3 with the reference drug in terms of features related to its molecular mechanism of action was evaluated. For the extracellular effects of SB3, cancer cells overexpressing HER2 were used to evaluate HER2 receptor expression, the extracellular domain of HER2, and the activity of antibody-dependent phagocytosis. For intracellular effects, Akt phosphorylation and vascular endothelial growth factor release were evaluated. The authors also performed in vitro tests using a combination of docetaxel or pertuzumab. Anti-proliferation, HER2/HER3 dimerism inhibition, apoptosis, and antibody-dependent cytotoxicity assays were used. The results confirmed that SB3 is similar to the reference product in quality characteristics related to both extracellular and/or intracellular efficacy. The similarity was also confirmed by studies in combination with docetaxel and pertuzumab [22].

Dual blockade with trastuzumab and pertuzumab in combination with neoadjuvant chemotherapy is increasingly used in patients with HER2+ tumors larger than 2 cm and/or axillary positive lymph nodes to evaluate the histological response and apply the more relevant treatment. SB3 demonstrated similar efficacy in adjuvant chemotherapy with trastuzumab. Berg et al. used the Danish BC group database to evaluate outcomes in 215 patients receiving adjuvant chemotherapy with cyclophosphamide and epirubicin followed by weekly administration of paclitaxel or other chemotherapy followed by paclitaxel in combination with SB3. The results showed a complete histological response (CHR) rate (primary endpoint) of 56%. In addition, in 60 of the 88 patients with positive lymph nodes, ypN0 (i-) was achieved after neoadjuvant therapy. The rate of CHR was significantly correlated with estrogen receptors and the degree of malignancy [23].

In a phase III study, Pivot et al. compared SB3 with reference trastuzumab in 800 patients with HER2+ EBC. Patients were randomized to a reference SB3 or trastuzumab for eight cycles simultaneously with chemotherapy (four cycles of docetaxel followed by four cycles of fluorouracil, epirubicin, and cyclophosphamide and surgery followed by 10 cycles of SB3 or trastuzumab). The primary endpoint of the study was the rate of CHR, while the secondary endpoints were comparisons with the reference drug for the total rate of CHR, disease-free interval, OS, safety, pharmacokinetics, and immunogenicity. The rates of CHR were 51.7% and 42.0% with reference SB3 and trastuzumab, respectively. The overall CHR rates were 45.8% and 35.8%, while the OR rates were 96.3% and 91.2% with SB3 and trastuzumab, respectively [24].

##### Side Effects

Regarding the safety of SB3, Pivot et al. evaluated its long-term cardiac safety and efficacy in patients with HER2+ EBC treated with biosimilar SB3 or trastuzumab in a phase III study. The incidence of symptomatic congestive heart failure, the asymptomatic reduction in the left ventricular ejection fraction, the incidence of other cardiac events, survival without events, and OS in 367 patients (186 received SB3 and 181 trastuzumab) were monitored. During the two-year follow-up after adjuvant therapy, the incidence of asymptomatic reduction in the left ventricular ejection fraction was very low (SB3 = 1, trastuzumab = 2), and no cases of symptomatic heart failure or other cardiac events were reported. At the three-year follow-up, the event-free interval was 91.9% for SB3 and 85.2% for trastuzumab. The percentage of patients with side effects was 9.1% with SB3 and 17.1% with trastuzumab, respectively [25]. In the phase III study of Pivot et al., 96.6% and 95.2% of patients had one or more side effects, 10.5% and 10.7% had serious side effects, and 0.7% and 0.0% showed antibodies to the drug with SB3 and reference trastuzumab, respectively. These data suggest that there is equivalence between SB3 and reference trastuzumab with comparable safety and immunogenicity parameters [24].

#### 3.1.2. CT-P6 (Trastuzumab-Pkrb, Herzuma)

The biosimilar CT-P6 (Herzuma^®^; Celltrion Inc., Incheon, Korea) is connected with high specificity and related to the same HER2 epitope as the Herceptin^®^ (Genentech) reference product. It was approved in the USA for the treatment of HER2+ EBC and MBC in the year 2018 [26,27].

Herzuma has been studied as neoadjuvant therapy in patients with HER2+ EBC. The existing data suggest that CT-P6 has similar safety and efficacy profiles to the reference drug trastuzumab. Both preclinical and clinical studies have shown equivalence between CT-P6 and reference trastuzumab. The preclinical pharmacological model of CT-P6 shows a similar mechanism of action, similar pharmacological properties, and similar pharmacokinetics to reference trastuzumab. A multicenter randomized phase III clinical trial in patients with early BC demonstrated equal safety and efficacy between CT-P6 and trastuzumab. Furthermore, the one-year follow-up of patients showed the same rates of cardiotoxicity. CT-P6 is a safe and effective alternative for use in patients with HER2+ BC. Its application in clinical practice will increase patients’ access to modern treatment modalities and will offer financial relief to healthcare systems [26].

The primary objective of the study by Esteva et al. was to evaluate and compare the properties of the biosimilar CT-P6 and the reference trastuzumab in healthy subjects. The secondary objectives were to compare the safety and immunogenicity of CT-P6 and reference trastuzumab. A single-dose, randomized, double-blind, and parallel-group study comparing CT-P6 with reference trastuzumab (6 mg/kg, 90-min intravenous infusion) was performed on 70 healthy adult men. Pharmacokinetics, safety, and immunogenicity were assessed 10 weeks after dosing. Equivalence of CT-P6 and the reference trastuzumab was demonstrated. The ratios (CT-P6/trastuzumab reference) of the LS geometric means were: AUCinf 99.05 (CI = 93.00–105.51); AUClast 99.30 (92.85–106.20); Cmax 96.58 (90.93–102.59). The safety profiles were similar. Adverse reactions occurred in 10 subjects (28.6%) in the CT-P6 group and 11 subjects (31.4%) in the trastuzumab control group. There were no serious side effects or deaths. No subjects were found to be positive for antibodies to these drugs [28].

In a randomized, double-blind, Phase III equivalence study, women aged 18 years and older suitable for surgical treatment for HER2+ BC stage I-IIIa from 112 centers in 23 countries were recruited. Patients were divided into a 1: 1 ratio to receive adjuvant therapy CT-P6 or trastuzumab intravenously (eight cycles, lasting 3 weeks for 24 weeks, 8 mg/kg BW on day 1 of cycle 1, and 6 mg/kg BW on day 1 of cycles 2–8) in combination with neoadjuvant therapy with docetaxel (75 mg/m^2^ on day 1 of cycles 1–4) and fluorouracil (500 mg/m^2^), epirubicin (75 mg/m^2^), and cyclophosphamide (500 mg/m^2^) (day 1 of cycles 5–8). Surgery was performed within 3–6 weeks of the final dose of the neoadjuvant drug followed by an adjunctive treatment period of up to one year. The safety and efficacy of treatment were evaluated in 549 patients for 3 years (271 on CT-P6 and 278 on reference trastuzumab). A complete response rate was observed in 116 of 248 patients (46.8%) of the CT-P6 group and 129 of 256 patients (50.4%) of the trastuzumab control group. In the CT-P6 group, 19 of the 271 patients (7%) reported severe adverse reactions compared with 22 of 278 (8%) in the trastuzumab control group. The most common serious side effect was febrile neutropenia (4 (1%) vs. 1 (<1%)) and neutropenia (1 (<1%) vs. 2 (1%)). Treatment-related adverse reactions occurred in 17 (6%) of the 271 patients in the CT-P6 group vs. 23 (8%) of the 278 patients in the trastuzumab control group. The most commonly observed adverse reaction was neutropenia in 10 (4%) vs. 14 patients (5%) [27].

Jeong et al. investigated the possibility that CT-P6 exerts its effects through the same mechanism of action as trastuzumab. The authors compared the mechanism of action of CT-P6 and trastuzumab both as monotherapy and in combination with paclitaxel or pertuzumab in models of BC and gastric cancer cell lines overexpressing HER2. The results confirmed the initial hypothesis that CT-P6 acts like trastuzumab by binding to the HER2 receptor [29].

Bae et al. [30] demonstrated that CT-P6 has similar efficacy and cardiac safety to trastuzumab reference in HER2+ EBC and MBC patients when administered as part of dual-target HER2 therapy with pertuzumab in combination with neoadjuvant or adjuvant chemotherapy. The authors retrospectively evaluated 254 patients with HER2+ EBC who had received adjuvant chemotherapy with trastuzumab reference or CT-P6, along with pertuzumab, carboplatin, and docetaxel (TCHP), and 103 patients with stage IV MBC who had previously been submitted to palliative treatment with trastuzumab reference or CT-P6 plus pertuzumab and docetaxel (THP) between May 2014 and December 2019. The primary endpoints of the study were a CHR in EBC and free disease progression (FDP) in the MBC cohort, while OS OR rate, disease control rate, and cardiac outcomes were the secondary endpoints. The authors found that a similar proportion of patients with EBC achieved a CHR on CT-P6 therapy compared with reference trastuzumab (74.4% (93/125) vs. 69.8% (90/129), *p* = 0.411). In patients with MBC, the median follow-up was 23.0 and 41.0 months for the reference CT-P6 and trastuzumab groups, respectively. DFS did not differ significantly between the two groups (13.0 vs. 18.0 months, 95% CIs 0.0–26.6 vs. 11.3–24.7). Cardiac events also did not show a significant difference between the two groups.

##### Side Effects

The side effects of CT-P6 are considered to be similar to those of the reference drug. In a related study, and after adjuvant chemotherapy and surgery, Esteva et al. administered adjuvant therapy with CT-P6 (271 patients) or trastuzumab (278 patients, 6 mg/kg/3 weeks) for at least one year. The results showed that CHR rates (tpCR) and complete breast histological response rates (bpCR) were comparable in the two groups regardless of age or clinical stage. In total, 47.6% in the CT-P6 group and 52.2% in the trastuzumab group experienced treatment-related adverse reactions. These percentages include 17 patients who reported heart failure (CT-P6: 10, trastuzumab: 7). Two patients in the CT-P6 group and three patients in the trastuzumab group discontinued adjuvant therapy due to side effects [31].

In a recent study, Stebbing et al. [32] described data on the safety and efficacy of CT-P6 and trastuzumab in patients with HER2+ EBC after at least 3 years of follow-up. Patients after adjuvant chemotherapy with CT-P6/trastuzumab underwent surgery and continued to receive adjuvant therapy with biosimilar CT-P6 (259) or trastuzumab (269 patients). Several parameters, including DFS and OS, cardiac adverse events, and immunogenicity, were assessed during the three-year follow-up. After a median follow-up of 38.7 months in the CT-P6 group and 39.6 months in the trastuzumab group, respectively, hazard ratios and three-year survival rates did not differ between groups. Similarly, data related to the safety of the two therapies were comparable between the groups for both the total study period and the follow-up period, including cardiac disorders (CT-P6: 22 (8.1%) patients vs. trastuzumab: 24 patients (8.6%)) and reduction in the left ventricular ejection fraction. Immunogenicity was similar between groups.

Regarding the total time of intravenous administration of the drug, it is recommended that CT-P6 be administered 30–90 min in maintenance infusions to prevent infusion-related reactions. Saito et al. [33] evaluated the safety of administering CT-P6 for 30 min as the first injection and alternatively to trastuzumab reference in maintenance infusions. A total of 140 patients with breast or gastric cancer who changed their three-week reference trastuzumab with CT-P6 to 30 min in maintenance infusions were evaluated retrospectively. The main endpoint was the effect of the reactions that occur during the infusion. The secondary endpoints were the incidence of diarrhea and dermal toxicity. Of these patients, 95% had breast cancer, and 44.3% had advanced stage cancer. Treatment included CT-P6 alone (17.9%) or in combination with cytotoxic agents (23.6%), antihormonal drugs (25.7%), and pertuzumab (62.9%). One patient (0.7%) had grade 3 infusion-related reactions. The incidence of diarrhea in the trastuzumab control group and the CT-P6 group was 7.1 and 6.4%, respectively, and that of skin toxicity was 6.4 and 5.0%, respectively.

#### 3.1.3. ABP 980 (KANJINTI™)

ABP 980 (KANJINTI ™, Amgen, Thousand Oaks, CA, USA; Amgen Europe B.V., The Netherlands) is a monoclonal antibody biosimilar to trastuzumab (Herceptin^®^, Genentech, South San Francisco, CA, USA; Roche Registration GmbH, Grenzach-Wyhlen, Germany) that acts against the HEP2 receptor [34]. ABP 980 is approved in the USA, EU, and Japan for all indications for trastuzumab reference products including HER2+ BC, HER2+ MBC, and gastroesophageal carcinoma. Its analytical characteristics are similar to those of the reference product, e.g., binding to the HER2 receptor inhibiting its activation and the proliferation of HER-expressing cells [35]. It is available as a lyophilized powder just like the reference product, dissolved and reconstituted before intravenous administration.

All the evidence suggests that ABP 980 is similar to the trastuzumab reference product in terms of physicochemical characteristics, such as primary structure, amino acid sequence, protein content, and content of drug-related substances [36]. In a study involving healthy volunteers, the geometric mean and reliability intervals (90%) of the pharmacokinetic parameters of the two products (AUC0-inf and Cmax) were within acceptable limits (0.80–1.25). ABP 980 has the same amino acid sequence and molecular mass as the reference trastuzumab. The comparative study of the glucan content of ABP 980 and trastuzumab showed that, although the levels of some glycans differed, the levels of essential glycans, which are associated with the ability to bind to the FcγRIIIa receptor, thus affecting antibody-dependent cytotoxicity, were similar. The assessment of functional similarity between ABP 980 and trastuzumab assessed with Fab-, Fc-, and combined Fab- and Fc-mediated activity, was similar in both drugs. ABP 980 has shown a similar ability to bind to the extracellular portion of HER2, similar potency, and similar antibody-dependent cytotoxicity in experimental systems based on cellular models [37,38].

A study also showed that ABP 980 and reference trastuzumab had similar on-rate, off-rate, and extracellular affinity for the extracellular domain of the HER2 receptor, as calculated by surface plasmon resonance for all ABP 980 values to be included on the verge of trastuzumab. In a BC SK-BR-3 cell binding study, the mean HER2 cell binding of all products tested was approximately 100%, thus confirming the similar activity of both drugs. Post-binding of HER2’s internalization confirmed the similar levels of internalization achieved with both drugs. Similar inhibition of proliferation with ABP 980 and trastuzumab in NCI-N87 gastric cancer cells expressing high levels of HER2 and a similar lack of inhibition of proliferation in BC MCF7 cells expressing low levels of HER2 were confirmed. Finally, a synergistic effect was demonstrated with docetaxel co-administration in studies using gastric cancer cells in vitro [39,40].

ABP 980 was also clinically evaluated in a randomized study in healthy adults. The study confirmed the absence of differences between the two drugs [41]. A comparative study in patients with EBC showed a similar rate of histologically complete response, as well as similar results regarding the safety and tolerance of the two antibodies. The rate of side effects and immunogenicity did not differ. Finally, the clinical efficacy and safety of ABP 980 were demonstrated in the multicenter, randomized, double-blind study (LILAC) in patients with EBC [63 [42]. In another study, 725 patients were randomized to receive ABP 980 (*n* = 364) or reference trastuzumab (*n* = 361) with paclitaxel (175 mg/m^2^ every 3 weeks or 80 mg/m^2^ weekly for 12 cycles), followed by surgery. CHR was achieved in 172 of 358 patients (48%) in the ABP 980 group and 137 of 338 patients (41%) in the trastuzumab control group. CHR was achieved in 162 of 339 (48%) patients in the ABP 980 group (at baseline) and in 138 of 330 (42%) patients in the trastuzumab reference group (at baseline) in a central laboratory evaluation [43,44].

##### Side Effects

In the LILAC study, patients receiving adjuvant therapy and surgery were randomized to receive adjuvant therapy with ABP 980 or reference trastuzumab every 3 weeks for 1 year from the start of treatment. During the adjuvant phase of the study, patients who had previously received ABP 980 continued with ABP 980 (6 mg/kg), while patients who had previously received trastuzumab continued to receive trastuzumab or ABP 980. Overall, the type, frequency, and severity of adverse reactions were similar between ABP 980 and reference trastuzumab in both phases of the study [43]. Furthermore, there was no difference in the incidence of adverse reactions between patients who switched from the reference trastuzumab to ABP 980 and those who continued to receive trastuzumab in the adjuvant environment. The left ventricular ejection fraction did not change in any treatment group during the study, and the change had no positive or negative effect. The overall incidence of heart disorders was low throughout the study. There were also no differences in the frequency of side effects between the group that changed from trastuzumab to ABP 980 and the one that continued to receive trastuzumab. The left ventricular ejection fraction did not change during the study [44].

Jassem et al. evaluated in vitro and in vivo the pharmacological similarities of ABP 980 with the reference trastuzumab (analytical and functional evaluation). This study completes the functional similarity assessment with additional in vitro studies and non-clinical studies including HER2 receptor binding, HER2 “internalization”, binding with the natural killer and monocyte cells, antigen-dependent cell phagocytosis, in vivo xenograft studies, and toxicokinetic parameters. The results demonstrate the functional similarity of the two antibodies, suggesting that ABP 980 is similar to trastuzumab in all primary and secondary mechanisms of action [38].

In the LILAC phase III study, ABP 980 showed similar clinical efficacy and tolerability to the reference trastuzumab in patients with HER2+ EBC. The tolerance, immunogenicity, and safety profiles of ABP 980 were similar to those of trastuzumab. The change from reference trastuzumab to ABP 980 did not affect the immunogenicity or safety of ABP 980 [45].

von Minckwitz et al. [43], compared the clinical safety and efficacy of ABP 980 with that of trastuzumab in women with HER2+ EBC in a randomized, multicenter, double-blind, controlled trial conducted in 97 centers in 20 countries. After four cycles of anthracycline-based chemotherapy, patients received ABP 980 or reference trastuzumab. Patients received neoadjuvant therapy at a dose of 8 mg/kg ABP 980 or trastuzumab and paclitaxel 175 mg/m^2^ in a 90 min intravenous infusion, followed by three cycles of 6 mg/kg intravenous ABP 980 or paclitaxel 7 mg/kg/m^2^ every 3 weeks in 30 min intravenous infusions. Surgery was performed 3–7 weeks after the last dose of neoadjuvant therapy. Following surgery, adjuvant therapy with ABP 980 or reference trastuzumab was given every 3 weeks for 1 year. Primary efficacy endpoints were risk difference and risk ratio of CHR in the breast and axillary lymph nodes. The safety of the drugs in all patients who received any amount of the product under investigation was also studied. Finally, 725 patients were randomized to receive ABP 980 (*n* = 364) or reference trastuzumab (*n* = 361). CHR was found in 172 (48%) of 358 patients in the ABP 980 group and 137 patients (41%) of the 338 patients in the trastuzumab control group (90% CI 1033–1366). CHR (central laboratory evaluation) was observed in 162 (48%) of the 339 patients treated with ABP 980 and in 138 (42%) of the 330 patients treated with trastuzumab (90% CI 0.993 to 1.312). Adverse reactions during the neonatal therapy phase occurred in 54 (15%) of the 364 patients in the ABP 980 group and 51 (14%) of the 361 patients in the trastuzumab group, the most significant of which was neutropenia (6% in both groups). In the adjuvant phase, grade 3 adverse reactions occurred in 30 (9%) of the 349 patients who continued treatment with ABP 980, 11 (6%) of the 171 who continued to receive reference trastuzumab, and 13 (8%) of the 171 who changed from reference trastuzumab to ABP 980. The most common side effects were infections or parasitoses (1%), neutropenia (1%), and infusion reactions (1–2%) in all groups. Two patients died from side effects not related to the treatment. One patient died of pneumonia and one from septic shock. Based on the central laboratory evaluation of the tumor samples, the results suggest similar efficacy of both drugs. ABP 980 and reference trastuzumab did not differ in safety parameters in either the neoadjuvant or adjuvant phases of the study.

#### 3.1.4. PF-05280014 (Trazimera^TM^)

PF-05280014 (Trazimera™) is the fourth biosimilar to the anti-HER2 trastuzumab reference antibody approved in the EU for use in all cases for which the reference trastuzumab has been approved, including HER2+ MBC and HER2+ EBC. PF-05280014 has similar physicochemical and pharmacodynamic properties to those of the trastuzumab reference drug. The pharmacokinetic similarity of the two drugs has been demonstrated in women with BC and in healthy male volunteers. The efficacy of PF-05280014 was found to be equivalent to that of trastuzumab reference in women with HER2+ MBC and comparable to that of trastuzumab reference in women with HER2+ EBC. The immunogenicity, tolerability, and safety profiles of PF-05280014 were similar to those of reference trastuzumab [46].

Chen et al. [45] compared the efficacy, safety, and immunogenicity of PF-05280014 with those of trastuzumab of EU origin (trastuzumab-EU) in combination with paclitaxel in a total of 702 patients with HER2+ BC (PF-05280014: 349 patients, trastuzumab-EU: 353 patients). PF-05280014 and trastuzumab-EU had similar pharmacokinetic parameters in patients with HER2+ MBC.

In a randomized, double-blind study, Li et al. [47] compared the biosimilar trastuzumab PF-05280014 (352 patients) with the reference trastuzumab (Herceptin^®^) from the EU (trastuzumab-EU) (355 patients) with concurrent administration of paclitaxel as a first-line treatment for HER2+ MBC. PF-05280014 and trastuzumab-EU were administered in weekly doses (first dose 4 mg/kg, subsequent doses 2 mg/kg), with a 3-week regimen (6 mg/kg) from week 33. Paclitaxel (80 mg/m^2^) was administered on days 1, 8, and 15 of the 28-day cycles for at least six cycles. The main endpoint was the objective response rate (ORR). A total of 451 (63.8%) patients completed the study. The risk ratio for the ORR was found to be 0.940 (62.5% for the PF-05280014 group and 66.5% for the trastuzumab-EU group). The time to stop treatment did not differ significantly between the two groups (12.2 vs. 12.0 months, respectively). Death rates during follow-up also did not differ significantly (17.3 vs. 18.9%). Survival rates at two years were 82.3 vs. 77.4%, while at 3 years, they were 77.2 vs. 75.3%. The frequency of adverse reactions (98.6 vs. 96.6%) did not significantly differ between the two groups. The immunogenicity results were similar between the two groups.

In a randomized, double-blind study, Lammers et al. compared the pharmacokinetics, efficacy, safety, and immunogenicity of PF-05280014 and the reference product trastuzumab (Herceptin) from the EU (trastuzumab-EU) in patients with operable HER2+ BC. A total of 226 patients stratified based on primary tumor size and hormone receptor status were randomized 1:1 to receive either PF-05280014 or trastuzumab-EU (8 mg/kg loading dose, 6 mg/kg subsequently) in parallel with docetaxel and carboplatin every 3 weeks for a total of six treatment cycles. The primary endpoint was the percentage of patients with Concentration trough plasma levels (Ctrough) greater than 20 μg/mL in cycle 5. Efficacy endpoints included the complete histological response as well as the rate of objective response. Similar percentages were found in the two groups (92.1% vs. 93.3%), with trough levels in cycle 5 greater than 20 μg/mL. The rates of complete histological response (47.0% vs. 50.0%) and the rates of objective response (88.1% vs. 82.0%) were similar. The incidence of adverse reactions during treatment was 38.1% vs. 45.5%, respectively. Antidrug antibody rates were 0% vs. 0.89%. It therefore appears that PF-05280014 has comparable pharmacokinetics, efficacy, safety, and immunogenicity to trastuzumab-EU in patients with operable HER2+ BC receiving adjuvant chemotherapy [48].

#### 3.1.5. Trastuzumab Dkst (OGIVRI)

The trastuzumab OGIVRI biosimilar was the first one approved by the US FDA for patients with HER2+ BC and stomach cancer. HERITAGE is a multicenter, double-blind, randomized study in which patients were randomized to receive either the biosimilar trastuzumab-dkst or trastuzumab and taxane followed by continuous monotherapy. Overall survival was assessed either 36 months after the start of the study or after 240 deaths since the last patient was randomized. In the final analysis (36 months), 242 patients had died during the study (116 and 124 in the trastuzumab-dkst and trastuzumab control groups, respectively). The median overall survival was 35 months with trastuzumab-dkst and 30 months with reference trastuzumab. The disease-free survival period did not differ between the two groups (11.1 months). No adverse events were reported. Therefore, this first phase III study, which looked at long-term survival after treatment with biosimilar trastuzumab, found long-term survival similar to the reference trastuzumab in patients with MBC and identified the biosimilar trastuzumab-dkst as similar to the reference trastuzumab in terms of clinical outcomes in patients with HER2+ BC [49].

Table 5 summarizes the main safety and efficacy characteristics and results of studies performed on HER2-positive BC patients, as discussed above.

**Table 5 biomedicines-10-02045-t005:** Summary of clinical trials evaluating the safety and efficacy of approved trastuzumab biosimilars in patients with HER2+ breast cancer.

Ref.	Chemotherapy Regimens	Cohort Size	Phase	Settings	Endpoints								
					%C_trough_, μg/mL	%ORR	%EFS	PFS, %HR	OS, %HR	%bpCR	DFS, %HR	TTD, mo	AEs/SAEs
[24] Pivot et al., 2018	Neoadjuvant phase (8 cycles): SB3 (*n* = 402) vs. TRZ (*n* = 398) plus chemotherapy (docetaxel -> FEC) Adjuvant phase (10 cycles) of SB3 or TRZ as monotherapy	800	III	Patients with HER2+ early breast cancer in the neoadjuvant setting	96.3% vs. 91.2%				51.7% vs. 42.0%		45.8% vs. 35.8 %		96.6% (SB3) vs. 95.2% (TRZ) of patients experienced one or more AEs SAEs: 10.5% (SB3 group) vs. 10.7% (TRZ group) 0.7% and 0.0% had antidrug antibodies (up to cycle 9) with SB3 and TRZ, respectively
[25] Pivot et al., 2019	SB3 (*n* = 186) vs. TRZ (*n* = 181)	367	III (follow-up)	Patients who completed the phase 3 study		(at 3 years): 91.9% with SB3 and 85.2% with TRZ							Total AEs: 9.1% with SB3 and 17.1% with TRZ Incidence of asymptomatic significant LVEF decrease was rare (SB3, *n* = 1; TRZ, *n* = 2) No cases of symptomatic CHF or other cardiac events were reported
[27] Stebbing et al., 2017	Neoadjuvant CT-P6 (*n* = 271) vs. TRZ (*n* = 278) plus docetaxel (Cycles 1–4) ->FEC (cycles 5–8)	549	III	Women ≥ 18 yrs with stage I–IIIa operable HER2+ breast cancer						46.8% vs. 50.4%			Serious TEAEs: 7% in CT-P6 group vs. 8% in TRZ group (febrile neutropenia) Grade 3 or worse TEAES: 6% in the CT-P6 group vs. 8% of 278 in the TRZ group
[31] Esteva et al., 2019	CT-P6 (*n* = 271) vs. TRZ (*n* = 278)	549	III (post hoc analysis)	Female aged over 18 yrs with pathologically confirmed, newly diagnosed, operable HER2+ Breast cancer					Comparable between the two groups regardless of age, region, or clinical disease stage		Similar results with bpCR		Drug-related TEAEs: 47.6% (CT-P6) vs. 52.2% (TRZ)
[32] Stebbing et al., 2021	CT-P6 (*n*=259) vs. TRZ (*n*=269)	III (follow-up)	528	Post-treatment follow-up from Phase III equivalent study			1.31 (95% CI: 0.86–2.01)	1.10 (95% CI: 0.57–2.13)		1.23 (95% CI: 0.78–1.93)			Study drug-related cardiac disorders (CT-P6: 8.1% vs. TRZ: 8.6%) Immunogenicity was similar between groups
[43] von Minckwitz et al., 2018	Anthracycline-based chemotherapy-> ABP 980 (*n* = 364) or TRZ (*n* = 361)	III	725	Women ≥ 18 yrs with histologically confirmed HER2+ invasive early breast cancer, ECOG (0 or 1) in the neoadjuvant setting							48% vs. 41%		Neoadjuvant phase: Grade 3 or worse AEs: 15% vs. 14% (neoadjuvant phase), 9% and 6% (adjuvant phase) in the ABP 980 and TRZ group, respectively Neutropenia: 21 (6%) patients in both groups
[47] Li et al., 2022	PF-05280014 (*n* = 352) vs. TRZ (*n* = 355) plus paclitaxel	III	707	Women ≥ 18 yrs with metastatic, histologically confirmed HER2+ breast cancer (ECOG status: 0–2)					92.9% (95% CI: 0.656–1.316)			12.25 vs. 12.06	Incidences of TEAEs overall (98.6%; 96.6%) and for grades ≥3: 41.0%; 43.1%) in PF-05280014 and TRZ group, respectively
[48] Lammers et al., 2018	PF-05280014 (*n* = 114) vs. TRZ (*n* = 112)	III	226	Women older than 18 yrs with histologically confirmed HER2+ breast cancer	92.1% vs. 93.3%						47.0% vs. 50.0%		Grade 3–4 TEAEs: 38.1% vs. 45.5%, Antidrug antibody rates: 0% vs. 0.89%
[49] Rugo et al., 2021	Trastuzumab dkst (*n* = 230) vs. TRZ (*n* = 228) plus taxane -> continued monotherapy until disease progression	III	458 (ITT population) 343 (monotherapy/safety population	Patients with HER2+ mBC			35.7% vs. 37.7%	52.6% vs. 50.0%					Cumulative TEAEs and SAEs were similar in both groups, with few grade ≥ 3 TEAEs (69.3% vs. 72.6%), Immunogenicity was low and similar in both groups at 48 weeks

TRZ: trastuzumab, EFS: Event-Free Survival, PFS: Progression-Free Survival, DFS: Disease-Free Survival, TTD: Time to Treatment Discontinuation, bpCR: Breast Pathological Complete Response, FEC: Fluorouracil, Epirubicin, Cyclophosphamide, mBC: Metastatic Breast Cancer.

### 3.2. Biosimilars of Trastuzumab under Development

The trastuzumab biosimilars under clinical research and development are subsequently analyzed. A total of twelve studies are included, three of which were in phase III evaluating the efficacy and safety of a given biosimilar in patients with recurrent or metastatic HER2+ BC. The remaining nine studies were in phase I investigating the pharmacokinetic equivalence of biosimilars compared with reference trastuzumab in healthy volunteers.

#### 3.2.1. HLX02

Trastuzumab biosimilar HLX02 is manufactured in China and has been tested in clinical practice with phase III studies over the last two years. Zhu et al. [50], in a phase I study, evaluated the bioequivalence of HLX02 with the EU-derived trastuzumab in healthy adult males. The first part of the study evaluated the safety of different doses of the drug (2, 4, 6, or 8 mg/kg intravenous infusion for 90 min). The second part consisted of a randomized, double-blind study aiming to investigate the pharmacokinetics, safety, and immunogenicity of the study drugs (HLX02 (*n* = 37), CN-trastuzumab (*n* = 35), or EU-trastuzumab (*n* = 37)). The results concerning the first phase of the study showed that all doses of HLX02 were well-tolerated. The results of the second study using the dose of 6 mg/kg showed that the mean geometric ratio and the 90% Confidence Intervals (90% CIs) for the serum plasma concentration curve were 0.950, 0.914, and 0.962 for HLX02 vs. CN-trastuzumab, HLX02 vs. EU-trastuzumab, and CN-trastuzumab vs. EU-trastuzumab, respectively. Adverse reactions were reported in 75.7%, 86.5%, and 70.3% of subjects in the HLX02, CN-trastuzumab, and EU-trastuzumab groups, respectively. No serious side effects or deaths were reported. No antibodies to the drug were detected, suggesting similar safety and pharmacokinetic bioequivalence between HLX02, CN-trastuzumab, and EU-trastuzumab in healthy Chinese men.

In a recent study, Zhang et al. [51] evaluated the bioequivalence of the biosimilar HLX02 vs. the reference product (US-trastuzumab) in healthy Chinese men who received single doses of HLX02 (2–8 mg/kg). A randomized, double-blind, parallel study was then performed to investigate the similarities between HLX02 (6 mg/kg, 55 subjects) and US-trastuzumab (52 subjects). The pharmacokinetic properties of HL02 were found to be similar to those of US-trastuzumab. The bioavailability comparison with US-trastuzumab was similar in both drugs. A percentage of 81.7% in the HL02 group and 79% in the US-trastuzumab group experienced mild or moderate side effects. A slight increase in serum transaminases was the most common side effect.

In a randomized, double-blind phase III study, Xu et al. evaluated the efficacy, safety, and immunogenicity of HLX02 compared with reference trastuzumab in 649 patients with recurrent or HER2+ MBC. Patients received reference HLX02 or trastuzumab of EU origin (initial dose 8 mg/kg, followed by 6 mg/kg every 3 weeks for 12 months) in combination with docetaxel. The primary endpoint was the ORR at week 24. The ORRs were 71.3 and 71.4% in the HLX02 (*n* = 324) and EU-trastuzumab (*n* = 325) groups. No statistically significant differences were observed in any of the secondary endpoints. Safety and immunogenicity were comparable in the HLX02 and EU-trastuzumab groups. Overall, 98.8% of patients in each group experienced at least one adverse reaction. Of these, 23.8% and 24.9% had serious side effects, while 0.6% in each group developed antibodies to the drug. These results suggest that HLX02 exhibits equivalent efficacy to reference trastuzumab in patients with recurrent or HER2+ MBC, as well as similar safety and antigenicity [52].

#### 3.2.2. AryoTrust^®^ (AryoGen Pharmed Co., Iran)

AryoTrust^®^ (AryoGen Pharmed Co., Teheran, Iran) is a candidate biomarker of the EU reference trastuzumab (Herceptin^®^). In a double-blind, parallel study group, the bioequivalence of the AryoTrust^®^ and the reference trastuzumab were evaluated. Sixty healthy men were randomized to receive a single dose of AryoTrust^®^ or Herceptin^®^ (6 mg/kg) intravenously. The primary endpoint of the study was the area under concentration (AUC) compared to the time to infinity (AUC vs. time to infinity), while the secondary endpoints were immunogenicity, safety, and the maximum measured concentration (Cmax) area below concentration vs. time from zero. It was found that the pharmacokinetic parameters were similar for the primary and secondary endpoints (acceptance range of bioequivalence 80.0–125.0%). No serious adverse reactions or immunogenicity were reported. All of the side effects reported were mild and similar between the two treatment groups [53].

#### 3.2.3. SIBP-01

SIBP-01 is under investigation as a newly developed trastuzumab biosimilar. In a randomized, double-blind, phase I study, 100 healthy male volunteers were randomized to receive a single intravenous dose of 6mg/kg SIBP-01 or trastuzumab reference (Herceptin^®^). The geometric mean ratios of Cmax, AUC0-t, and AUCinf in the control and reference groups were found to be 93.5% vs. 104.27%, 91.98% vs. 102.3%, and 91.9% vs. 102.3%, respectively. In the sensitivity analysis, the geometric mean ratios of AUC0-t and AUCinf were 92.3% vs. 102.6% and 91.8% vs. 102.2%, respectively (being within the pre-defined equivalence limits). Side effects were reported at 72.0% and 80.0%, respectively. The protein molecular structure, biological activity, and purity of SIBP-01 are similar to those of Herceptin^®^. However, the absence of a sufficient number of studies examining the bioequivalence of SIBP-01 with Herceptin^®^ in healthy individuals and patients with BC should be seriously considered [54].

#### 3.2.4. TRASTUREL (Manufacturer: RELIANCE)

In a multicenter, randomized, phase III comparative study involving 42 patients with MBC, the safety and efficacy of a trastuzumab biosimilar (TRASTUREL) were compared with the reference trastuzumab. A loading dose of 8 mg/kg trastuzumab was administered intravenously on day 1 of the first cycle. Serum samples were taken at predetermined intervals. Cmax and AUC0-336 were calculated for a single dose. A total of 106 patients were randomly assigned to receive biosimilar trastuzumab or reference trastuzumab with paclitaxel. The primary endpoint was the ORR. Secondary endpoints included the time to tumor progression, progression-free survival, total survival at week 48, and safety parameters. For the biosimilar and the reference trastuzumab, the mean Cmax was 229.0 and 210.7 μg/mL, respectively, while the AUC0-336 was 24,298.3 and 25,809.3 (μg × h/mL), respectively. Biosimilar trastuzumab and reference trastuzumab had a similar ORR (48.5% vs. 44.5%). Complete and partial response rates were similar in both groups. Side effects occurred in 68.3% and 59.1% of the biosimilar and the reference drug, respectively. No antibody development was observed against either drug in any of the patients in the study [55].

#### 3.2.5. AK-HER2

AK-HER2, a recombinant human monoclonal antibody to HER2, is a potential biosimilar to the reference drug Herceptin^®^. Wang et al. [56], in a randomized, double-blind, phase I study, investigated the pharmacokinetic equivalence between the biomarker under investigation and the reference trastuzumab, as well as their safety and immunogenicity, in 96 healthy volunteers who received an intravenous AK-HER2 or trastuzumab at a dose of 6 mg/kg. Primary endpoints were the AUC from time 0 to the last time point (AUC0-t) as well as the Cmax. The results showed that the pharmacokinetic parameters of the two drugs were similar. The Cmax and AUC0-t of women in the biosimilar group were higher than those of men (*p* < 0.05). No infusion-related reactions or positivity to antibodies to the drug were observed.

#### 3.2.6. BAT8001 (Bio-Thera Solutions, Guangzhou, China)

BAT8001 is an antibody conjugate with Batansine against HER2. Hong et al. [57] evaluated the safety, tolerability, pharmacokinetics, and antineoplastic activity of BAT8001 in 29 patients with locally advanced or HER2+ MBC in a phase I study. Patients received BAT8001 intravenously over a 21-day cycle, with dose escalation in five cohorts: 1.2, 2.4, 3.6, 4.8, and 6.0 mg/kg. The primary endpoint was the safety and tolerability of the BAT8001. The results showed that the toxic effects that necessitated dose reduction were thrombocytopenia and elevated transaminases. The maximum tolerated dose was found to be 3.6 mg/kg. Significant side effects occurred in 48.3% of patients, of which thrombocytopenia occurred in 12 (41.4%), an increase in aspartate aminotransferase in 4 (13.8%), an increase in γGT in 2 (6, 9%), increase in ALT in 2 (6.9%), and diarrhea in 2 (6.9%) patients. Objective response was observed in 12 (41.4%) and disease control (including patients who achieved objective response and stable disease) in 24 (82.8%) patients.

#### 3.2.7. BCD-022 (Herticad^®^, JSC BIOCAD, Saint Petersburg, Russia)

BCD-022 is a biosimilar of trastuzumab that is equivalent to the reference trastuzumab in a wide range of physicochemical studies, as well as in in vitro and in vivo preclinical studies. The efficacy and safety of BCD-022 and reference trastuzumab in combination with paclitaxel were evaluated in a multicenter phase III clinical trial. A total number of 225 patients with HER2+ MBC were randomized to receive a 1:1 ratio of BCD-022 or the reference trastuzumab with paclitaxel. Treatment was continued for 6 cycles of 3 weeks until disease progression or severe toxicity. The primary endpoint was the ORR. The results showed that the ORR was 49.6% in the BCD-022 and 43.6% in the trastuzumab reference arm. The profile of adverse reactions was similar between groups (93.8% in the BCD-022 arm and 94.5% of patients in the control arm). No adverse reactions were reported throughout the study. In addition, no statistically significant differences were found in the incidence of antibodies development between BCD-022 (*n* = 3, 2.6%) and the reference drug (*n* = 4, 3.6%). Evaluation of pharmacokinetics after the first and sixth injections of the drug also showed equivalent parameters (AUC0-504, Сmах, Тmax, T1/2, C trough) [58].

#### 3.2.8. DMB-3111 (Meiji Seika Pharma, Tokyo, Japan)

DMB-3111 is a biosimilar to trastuzumab developed jointly by Meiji Seika Pharma (Japan) and Dong-A Socio Holdings (Korea). Morita et al. [59], in a phase I study, investigated the bioequivalence between DMB-3111 and trastuzumab and also compared the pharmacokinetics, safety, and immunogenicity of the two drugs in healthy adult Japanese men. A total of 70 individuals were randomized in a 1:1 ratio to receive either DMB-3111 or reference trastuzumab as a single intravenous infusion (6 mg/kg). It was noticed that DMB-3111 was bioequivalent to trastuzumab in terms of pharmacokinetics. The frequency of adverse reactions was also similar for both drugs.

#### 3.2.9. HD201 (Prestige Biopharma Ltd., Singapore)

HD201 is a recommended biosimilar product of trastuzumab, developed in Singa-pore by Prestige Biopharma Ltd. In the study of Pivot et al. [60], the pharmacokinetic parameters of HD201 were compared with the reference drug. Seventy-three subjects with similar anthropometric parameters were randomized to receive a single intravenous dose of 6 mg/kg HD201 or trastuzumab, respectively. The primary endpoint was the area below the AUC0-∞ serum concentration curve. Other pharmacokinetic parameters served as secondary endpoints. The study demonstrated the existence of pharmacokinetic equivalence between HD201 and trastuzumab. Regarding safety, the percentages of people who experienced drug-related side effects were 61.8% and 82.9% in the HD201 and EU-trastuzumab groups, respectively. The most commonly reported side effects were infusion reactions. No individual developed antibodies to the drug.

Demarchi et al. [61] also conducted a phase I study in healthy individuals to demonstrate pharmacokinetic equivalence. The main objective was to demonstrate the equivalence of HD201, EU-Herceptin^®^, and US-Herceptin^®^. An intravenous infusion of 6 mg/kg was administered to 105 healthy men. Parallel comparisons were made based on the primary endpoint (AUC0-inf) and the secondary endpoints (AUC0-last και Cmax). Overall, HD201 showed similarity in pharmacokinetic parameters to both EU-Herceptin^®^ and US-Herceptin^®^. The three drugs in the study also had similar safety profiles. The incidence of people with TEAEs of special interest was slightly lower in the HD201 group (20.0%) compared to the other treatment groups (EU-Herceptin^®^: 34.3%, US-Herceptin^®^: 31.4%). Only one person (EU-Herceptin^®^ group) developed antibodies to the drug before dosing.

Table 6 lists the most important clinical outcomes and safety data from phase III studies available for the three trastuzumab biosimilars under investigation.

**Table 6 biomedicines-10-02045-t006:** Summary of phase III clinical trials evaluating the safety, efficacy, and pharmacokinetics of trastuzumab biosimilars under investigation in patients with HER2+ breast cancer.

Ref.	Chemotherapy Regimens	Cohort Size	Phase	Settings	Endpoints						
					%ORR	AUC, μg × h/mL	PFS, mo	%PR	%CR	Cmax, μg/mL	AEs/SAEs
[52] Xu et al., 2021	HLX02 (*n* = 324) vs. TRZ (*n* =325) plus docetaxel	649	III (equivalence study)	Patients with HER2+ recurrent or metastatic BC	ORR_24_: 71.3 vs. 71.4%		11.7 vs. 10.6	66 vs. 67.7	5.2 vs. 3.7		Grade 3 or higher TEAEs: 85.8 vs. 86.4%; Serious TEAEs: 23.8 vs. 24.9%; Cardiac disorders of special interest: 4.9 vs. 5.2%
[55] Apsangikar et al., 2017	TRASTUREL vs. TRZ plus paclitaxel (1:1)	148	III	Patients with metastatic breast cancer	48.44% vs. 44.44%	24298.29 vs. 25809.33				229.02 vs. 210.68	AEs: 68.29%vs. 59.09%
[58] Alexeev et al., 2020	BCD-022 (115) vs. TRZ (110) plus paclitaxel	225	III	Female patients with no previous treatment for metastatic HER2(+) breast cancer	49.6% vs. 43.6%						AEs: 93.81% vs. 94.55%

TRZ: Trastuzumab, ORR: Objective Response Rate, PFS: Progression-Free Survival, PR: Partial Response, CR: Complete Response.

### 3.3. Cost–Benefit from the Use of Trastuzumab Biosimilars

Biosimilars offer significant savings in treatment costs with discounts of around 30% to 69% compared to reference products. Although the biosimilars market is still in its infancy, it is estimated that these factors reduced the total bio-related costs by USD 54 billion between 2017 and 2026 in the US [62,63]. In addition, physicians may be motivated to prescribe biosimilars vs. biological reference products. In some countries (e.g., Belgium and Germany), physicians are encouraged to prescribe biosimilars for a defined minimum number of patients. In other countries (e.g., Austria and Norway), doctors are required to prescribe the most economical or cheapest treatment option, which often involves biosimilars [64].

Two studies evaluated the potential cost savings of using biosimilar trastuzumab over reference trastuzumab. One study looked at the financial outcome of administering the biosimilar trastuzumab to Croatian patients who had not received trastuzumab for either MBC or EBC in 50% of patients. The projected savings in pharmaceutical expenditure ranged from EUR 0.26 million to EUR 0.69 million (with a hypothetical price reduction of 15% and 35%, respectively) after 1 year [65]. The second study “modeled” the financial implications of using trastuzumab biosimilars in 28 European countries and patients with HER2+ EBC, MBC, or metastatic gastric cancer [66]. With a hypothetical 30% discount, a drug change rate of 20% in the first year and 5% in the following years, the estimated savings in the first year will range from EUR 58 to 68 million. Over 5 years, the savings will range from EUR 0.91 billion to EUR 2.27 billion, which could allow 55,000 to 116,000 new patients to be treated with biosimilar trastuzumab.

Importantly, due to reduced costs, biosimilars are opening up new avenues of treatment for patients who previously could not afford the cost of biologic therapy [67]. The availability of biosimilars can also reduce the cost of reference products. The high cost of trastuzumab is a barrier to patients’ access to current treatment modalities. On the other hand, healthcare providers are reluctant to prescribe biologic therapies because of their high cost and despite their good efficacy. The availability of biosimilars could change this attitude. A 2014 study reported that almost half of oncologists would increase the prescription of anti-HER2 biological drugs if a lower-cost biosimilar may be available [68]. Cost reduction and better patient access to biosimilar therapies depend to a large extent on physician awareness and access to these life-saving medications [69]. Therefore, future cost-effectiveness analyses should include trastuzumab biosimilars and raise the awareness of healthcare providers about these emerging and cost-effective treatments. Finally, although several studies have estimated the cost savings of subcutaneous administration of trastuzumab compared to intravenous administration, these analyses did not take into account the benefits of using a trastuzumab biosimilar, which entered clinical practice after the expiration of the patent for the reference product in 2014 (EU) and 2019 (US) [70].

In a recent study, Cheng et al. [71] evaluated the cost-effectiveness of pertuzumab with biosimilar trastuzumab and docetaxel as initial treatment in patients with HER2+ MBC. They concluded that, although biosimilar trastuzumab reduced the cost of the pertuzumab combination regimen, the cost–benefit ratio remained high. Regarding the combined administration of biosimilars for HER2+ MBC, Diaby et al. showed that trastuzumab with docetaxel as the first line of treatment represented the most cost-effective strategy among the four regimens evaluated in patients with HER2+ MBC [72]. Genuino et al. [73], in a recent systematic review, evaluated the adjuvant administration of trastuzumab in comparison with chemotherapy alone in patients with HER2+ EBC. They analyzed data from 22 studies from high- and middle-income countries, respectively. The cost-effectiveness ratio in the high-income countries ranged from USD 6018 to USD 78,929 per year, with the corresponding range in the middle-income countries ranging from USD 3526 to USD 174,901. The results confirmed the satisfactory cost/effectiveness ratio of trastuzumab administration for HER2+ EBC. It should be emphasized that the scientific adequacy of the studies was high, while the quality of the data ranged from moderate to high. Finally, Al-Ziftawi et al., in a recently published systematic review of the cost-effectiveness of treatment in developing countries, evaluated 14 relevant studies published between 2009 and 2019. The majority of studies referred to the use of trastuzumab alone or in combination. In terms of the cost/effectiveness ratio, the results of all these studies showed great variation. The conclusion of this systematic review was that the use of anti-BC drugs in developing countries is not accompanied by a satisfactory cost/benefit ratio. More and more valid data from developing countries are needed for the evaluation of the pharmacoeconomic parameters of these newly approved drugs [74,75,76].

## 4. Discussion

Numerous scientific data support the assumption that the use of biological therapies has improved the survival expectancy as well as the quality of life of patients with serious diseases, including BC. At the same time, the use of these drugs offers significant economic benefits by reducing the cost of drug treatment and hospitalization [76]. However, despite their excellent clinical efficacy, the use of biological therapies is expensive, leading to a large number of patients not being able to receive these modern treatments. The aging of the population is a contributing factor, as it increases the number of chronically ill (with autoimmune or neoplastic diseases) patients requiring therapy with biological agents. The expiration of the exclusive patents of the first biopharmaceuticals and the growing demand have resulted in the development of biosimilars, i.e., biopharmaceuticals with high similarity to the original ones whose patent has expired. Biosimilars are less expensive, as they are free from the cost of primary research.

This systematic review dealing with the currently available biosimilars of the HER+BC trastuzumab reference product revealed that five biosimilars have been approved and are in use in various countries, including the USA and the EU, while nine others are under phase I and III evaluation. All studies were performed in comparison with the reference drug (trastuzumab) as far as the efficacy (primary and secondary endpoints) and the side effects (type and frequency) were concerned. It should be noted that the number of patients included in all studies was satisfactory, as was the methodology used to document and analyze the results. The vast majority of studies originated in Asia (China, Korea, and India), probably because in these countries, the need for biosimilar approval for HER2+ BC is urgent.

The review also showed that the clinical efficacy and safety of trastuzumab biosimilars were evaluated in a total of ten randomized, double-blind phase III studies on patients receiving adjuvant therapy or therapy for metastatic cancer. In all studies, the most common primary endpoint was the complete response rate. Other endpoints were disease-free survival, overall survival, and time to cessation of treatment. The results revealed that there were no statistically significant differences between the above-mentioned parameters, thus confirming the assumption that biosimilars are not inferior to the reference product. The results of all studies further showed that the frequency and type of side effects did not differ between biosimilars and the reference drug. The occurrence of cardiotoxicity as a major complication and milder side effects, including diarrhea, nausea, and vomiting, were not observed with increased frequency in the groups of patients who received the biosimilar compared to the reference trastuzumab. In addition, no development of antibodies to the biosimilar was observed, at least in a higher proportion as compared with the reference drug. In one study [43], neutropenia was observed in a small percentage of patients. In general, the incidence of side effects did not differ significantly between biosimilars and the reference drug. Therefore, the available results confirm that there are no obvious differences in clinical safety associated with the administration of biosimilars. There is also evidence from preclinical and clinical data that a combination of biosimilar and chemotherapy drugs or other biological therapies, such as pertuzumab, would have greater efficacy than the administration of trastuzumab.

Data from the general use of biosimilars in other situations are also interesting. For example, in 2017, the UK NHS saved USD 275 million from the use of infliximab, etanercept, and rituximab biosimilars, while the use of trastuzumab and adalimumab biosimilars in 2018 saved GBP 100 million. In the USA, the use of biosimilars could save USD 54 billion in 10 years [77]. The adoption of biosimilar drugs in Europe, in addition to reducing prices, is expected to facilitate patients’ access to specialized treatments. According to IMS Health (2016), the adoption of biosimilars in the markets of the EU and the USA would save USD 56–110 billion by 2020 [10]. The biosimilar market is constantly growing, as the need to increase access of patients to modern costly treatments is imposed by governments to save thousands of human lives, especially in developing countries.

Concerning the future of treatment of BC, it seems that the use of in vitro-transcribed mRNA (IVT-mRNA) aiming to induce the endogenous production of antibodies eliminates many of the previously described difficulties of antibody production. Rybacova et al. described an IVT-mRNA system for the in vivo production of trastuzumab to be used in the treatment of cancer [78]. They sequenced IVT-mRNA and then incorporated it into lipid nanoparticles to protect it from degradation so that it would be effectively released in vivo. This methodology resulted in the achievement of antibody concentrations in the serum of 45 ± 8.6 mg/mL in just 14 days after nanoparticle injection. Subsequent studies demonstrated a quite satisfactory pharmacokinetic profile of this antibody compared to reference trastuzumab. Furthermore, the administration of lipid nanoparticles to mice with cancer improved survival and reduced the tumor size. Therefore, it appears that the use of IVT-mRNA LNP for the in vivo generation of therapeutic antibodies may be an effective strategy for the treatment of BC as an alternative to trastuzumab administration.

## 5. Conclusions

This systematic review revealed that biosimilars—both those that have been approved and those under investigation—have the same efficacy and safety as trastuzumab. The pharmacological and clinical characteristics of both formulations are comparable. A large number of biosimilars have been approved so far, and many others are in the process of being tested for safety and efficacy. The result of this explosion in biosimilar growth is expected to save hundreds of thousands of lives worldwide each year, as an increasing number of patients will be able to access these modern treatments. It is encouraging that both the US FDA and Europe’s EMA are adopting biosimilars without going beyond the acceptable scientific safety standards for their approval. The development of trastuzumab biosimilars for subcutaneous administration (as is already the case with the reference medicine) will be a step forward in the treatment of patients, as a large number of them are expected to choose this route of administration. It is also expected that the hesitation of oncologists to choose a biosimilar instead of the reference drug will be eliminated shortly. Finally, the “conquest” of a large part of the drug market share by the biosimilars will lead to the reduction in pharmaceutical costs and the recovery of insurance systems, while incomparably more patients will have access to these treatments.

## Figures and Tables

**Figure 1 biomedicines-10-02045-f001:**
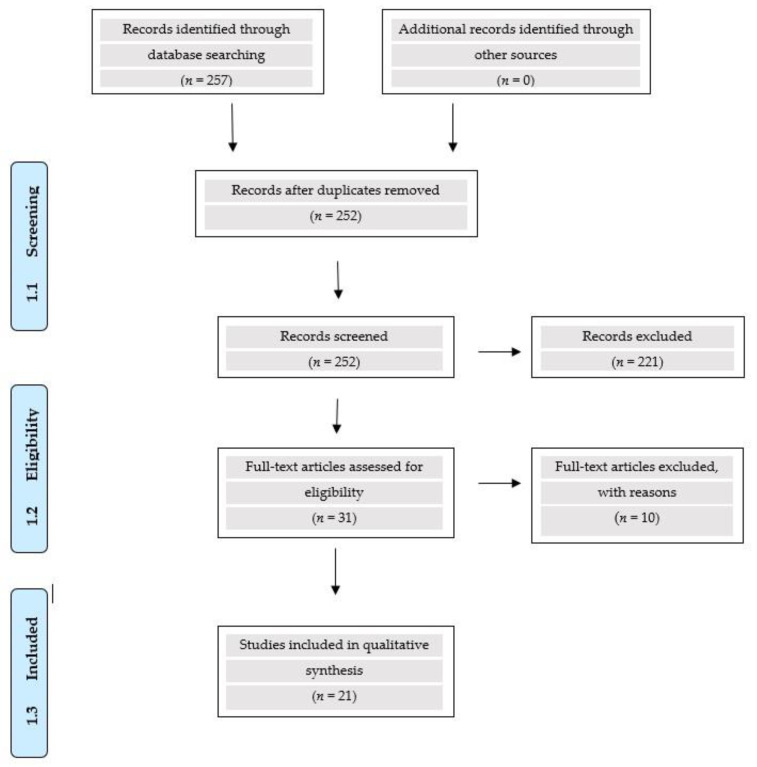
PRISMA Flow Diagram.

**Table 1 biomedicines-10-02045-t001:** Specific characteristics of biosimilar drugs (Amended by: *Biomaterials in the EU. Information guide for health professionals: European Medicines Agency and European Commission*).

A high degree of similarity with the reference drug	The biosimilar has physical, chemical, and biological properties that are very similar to the properties of the reference drug. There may be minor differences from the reference drug that are not clinically relevant for their safety or efficacy.
Absence of significant clinical differences compared with the reference drug	No clinically relevant differences are expected. Clinical studies supporting the approval of a biosimilar confirm that any differences do not affect both safety and efficacy.
Biodiversity of the biosimilar is maintained within strict limits	Minor variability is allowed only when scientific evidence indicates that it does not affect the safety and efficacy of the biosimilar. The permissible range of variability values for a biosimilar is the same as that allowed between the batches of the reference drug. This is achieved thanks to a thorough manufacturing process that ensures that all batches of the drug are of proven quality.
The same strict quality, safety, and efficacy standards are applied	Biosimilars are approved by the same strict quality, safety, and efficacy standards as any other pharmaceutical product.

**Table 2 biomedicines-10-02045-t002:** Definition of biosimilars according to international regulatory organizations [7].

Regulatory Authority	Regulatory Definition	Reference
The European Medicines Agency (EMA)	A biosimilar is a biological drug similar to another already approved “reference product”. The biosimilars are approved according to the same standards of pharmaceutical quality, safety, and efficacy that apply to all biological drugs.	The European Medicines Agency. Biosimilar Medicines
US Food and Drug Administration (US-FDA)	A biosimilar is a biological product very similar to a US licensed reference product, despite small differences in clinically inactive ingredients. There are no clinically significant differences between the biosimilar and the reference product in terms of safety, purity, and product efficacy.	42 U.S. Code § 262 (i) Regulation of biological products FDA Guidance for Industry: Questions and Answers regarding BPCIA (2015)
The Japanese Pharmaceuticals and Medical Devices Agency (Jp-PMDA)	A biosimilar is a new biotechnological pharmaceutical product developed in such a way that it makes it similar to an already licensed biotechnological pharmaceutical product (biological reference product). It has been developed based on data demonstrating comparability between the new biotechnological pharmaceutical product and the reference product in terms of quality, safety, and efficacy or other relevant data.	Japan Generic Medicines Association 25 November 2010 Interim Translation of Notification LED, PFSB, MHLW; Yakushokushinsa No. 0304007 (4 March 2009)
The World Health Organization (WHO)	The biosimilar is defined as “a biotherapeutic product, which is similar in quality, safety, and efficacy to an already licensed biotherapeutic reference product”.	WHO Guidelines on Evaluation of Similar Biotherapeutic Products (2009)

**Table 3 biomedicines-10-02045-t003:** Available and approved trastuzumab biosimilars.

Name	Biosimilar	Date of Approval (FDA)	Country, Manufacturer Company
Herxuma (trastuzumab-pkrb)	CT-P6	December 2018	Celltrion Inc., Incheon, Korea
Kanjinti (trastuzumab-anns)	ABP 980	June 2018	Amgen Inc., Thousand Oaks, CA, USA
Ogivri (trastuzumab-dkst)	MYL-1401O	December 2017	Mylan GmbH, Steinhausen, Switzerland
Ontruzant (trastuzumab-dttb)	SB3	January 2019	Samsung Bioepis Co., Ltd., Incheon, Korea
Trazimera (trastuzumab-pkrb)	PF-05280014	March 2019	Pfizer Ireland Pharmaceuticals, Ringaskiddy, Ireland

**Table 4 biomedicines-10-02045-t004:** Number of studies excluded with reasons for approved trastuzumab biosimilars.

Biosimilar	Number of Studies for Each Biosimilar	Phase Ι Studies	Phase ΙΙΙ Studies	Included in the Analysis	Excluded from the Analysis	Reasons for Exclusion
SB3 (ONTRUZANT)	4	1	3	1	3	Healthy volunteers were included; Updated data
CT-P6 (HERZUMA)	4	1	3	3	1	Healthy volunteers were included
ABP 980 (KANJINTI)	4	2	2	1	3	Healthy volunteers were included; Updated data
PF-05280014 (TRAZIMERA)	5	2	3	3	2	Healthy volunteers were included; Updated data
Trastuzumab dkst (OGIVRI)	2	0	2	1	1	Duplication
Total	19	6	13	9	10	

## Data Availability

Not applicable.

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
