# Peer review of "Systematic Review on the Use of Biosimilars of Trastuzumab in HER2+ Breast Cancer"

_biomedicines, 2022, doi:10.3390/biomedicines10082045_

Round 1
Reviewer 1 Report
The manuscript “Systematic Review on the Use of Biosimilars of Trastuzumab in Her-2+ Breast Cancer” falls within the scope of Journal. The findings are potentially interesting. However, this reviewer has the following comments for the manuscript:
-Authors should enlarge and improve the resolution of Table 5 and Table 6.
-Authors should report the keywords in alphabetical order.
-Authors should insert abbreviation section. The words for which is specified an abbreviation should be written in full the first time they are mentioned.
-The English language has to be extensively revised.
-Authors should improve the formal aspects of the manuscript.
-Authors should insert a Graphical abstract to summarizes the contents of the article in a concise form and capture the attention of the readership.
Author Response
The authors would like to thank the reviewer for his constructive remarks and comments on the article. His suggestions were taken into account and fully implemented. Relevant responses to specific comments and suggestions are as follows
Authors should enlarge and improve the resolution of Table 5 and Table 6.
Tables 5 and 6 were enlarged and their resolution improved
-Authors should report the keywords in alphabetical order.
Keywords were placed in alphabetical order
-Authors should insert abbreviation section. The words for which is specified an abbreviation should be written in full the first time they are mentioned.
An abbreviation section was created and introduced. The words were written in full the first time they were mentioned in the text.
-The English language has to be extensively revised.
The English text was extensively revised and submitted in a separate file. The authors have done their best to improve the English language of the text.
-Authors should improve the formal aspects of the manuscript.
The formal aspects of the manuscript were improved
-Authors should insert a Graphical abstract to summarize the contents of the article in a concise form and capture the attention of the readership.
A Graphical Abstract was created and included in the text.
Reviewer 2 Report
This systematic review by Triantafyllidi et al. aims to offer a comprehensive overview of Trastuzumab biosimilars. Trastuzumab is a blockbuster monoclonal antibody of extreme clinical significance. Therefore, I believe this review could be of high importance impact for the medical world to assess the available options.
I have only some minor comments for the authors to consider
Page 15 Table 5 and Page 19 Table 6 is not clear, please amend them with a new format so the readers can appreciate them, avoid the colors on the headlines also.
Lines 661-675 unbold them
Also efforts are ongoing to produce and test mRNA encoded Trastuzumab biosimilars see eg ref.
https://www.mdpi.com/1999-4923/13/9/1371
https://www.mdpi.com/2076-393X/9/8/890
Author Response
The authors would like to thank the reviewer for his kind comments on our work and his constructive remarks. His suggestions were taken into account and fully implemented. Relevant responses to specific comments and suggestions are as follows.
This systematic review by Triantafyllidi et al. aims to offer a comprehensive overview of Trastuzumab biosimilars. Trastuzumab is a blockbuster monoclonal antibody of extreme clinical significance. Therefore, I believe this review could be of high importance impact for the medical world to assess the available options.
Page 15 Table 5 and Page 19 Table 6 is not clear, please amend them with a new format so the readers can appreciate them, and avoid the colors on the headlines also.
Tables 5 and 6 were reformed and colors were removed
Lines 661-675 unbold them
Yes, thank you
Also efforts are ongoing to produce and test mRNA encoded Trastuzumab biosimilars see eg ref. [https://www.mdpi.com/1999-4923/13/9/1371 https://www.mdpi.com/2076-393X/9/8/890]
A special mention of the mRNA encoded trastuzumab was created and included in the text (discussion section) along with the relevant reference (81).